# Four in every ten infants in Northwest Ethiopia exposed to sub-optimal breastfeeding practice

**Workineh Shume Hailu**[1]**, Mulat Tirfie Bayih**[ORCID][2]*****, Netsanet Fentahun Babble**[2]

**1** Department of Midwifery, Mizan Aman Health Science College, Mizan Teferi, Ethiopia, **2** Department of Public Health Nutrition, School of Public Health, College of Medicine and Health Sciences, Bahir Dar University, Bahir Dar, Ethiopia

* mulatbonny@gmail.com

## Abstract

### Background

Improper breastfeeding practices significantly impair the health, development, and survival of infants, children, and mothers. A Breastfeeding Performance Index (BPI) is a composite index to describe overall maternal breastfeeding practice with infants under six months of age. To date, there is insufficient evidence on breastfeeding performance index and its associated factors in Ethiopia.

### Objective

To assess the breastfeeding performance index and its associated factors in Sekela District, Northwest Ethiopia, 2019.

### Methods

A community-based cross-sectional study was conducted on 605 randomly selected mothers having infants aged 6 to 12 months from April 02, 2019 to May 13, 2019. Data was collected using a structured interviewer-administered questionnaire. Multivariable logistic regressions were used to identify independent predictors of BPI.

### Results

Two hundred forty-six (40.7%) of mothers had low BPI scores. Mothers who lived alone (AOR = 3.18; 95%CI: 1.15, 8.82), mothers who were merchants (AOR = 2.75; 95%CI:1.05, 7.15), attended three antenatal care (ANC) visits (AOR = 0.42; 95% CI: 0.20, 0.82), attended four antenatal care visits (AOR = 0.35; 95%CI: 0.12, 0.82), received postnatal care (PNC) (AOR = 0.35; 95%CI: 0.19, 0.64), had poor knowledge on breastfeeding (AOR = 3.19;95%CI: 1.14, 8.89) or negative attitudes towards breastfeeding (AOR = 2.70;95%CI: 1.13, 6.45), were independent predictors of low BPI scores.

**Data Availability Statement:** All relevant data are within the manuscript and its Supporting Information files.

**Funding:** the authors received no funding for this work.

**Competing interests:** The authors have declared that no competing interests exist.

**Abbreviations:** ANC, Antenatal Care; BFPI, Breast Feeding Performance Index; EBF, Exclusive Breast Feeding; HEW, Health Extension Worker; IYCFP, Infant and Young Child Feeding practice; PNC, Post Natal Care; SPSS, Statistical Package for Social Sciences; TIBF, Timely Initiation of Breast Feeding; WHO, World Health Organization.

## Conclusions

The prevalence of sub-optimal breastfeeding practice in northwest Ethiopia was very high. A mother living alone, maternal occupation, ANC visits, PNC, maternal breastfeeding knowledge, and attitude towards breastfeeding were independent predictors of low BPI scores. Nutrition promotion should be implemented by considering the above significant factors to decrease inappropriate breastfeeding practice in Northwest Ethiopia.

## Introduction

Exclusive breastfeeding is recommended for the survival and healthy growth of the baby by providing the most nutritious food and transferring some of the mother's immunity to the infant. Breastfeeding creates an important psychosocial bond between the mother and baby and enhances modest cognitive development. It is the foundation of the infant's wellbeing for the first year to two years of a baby's life, when supplemental foods are withheld until after 6 months of exclusive breastfeeding [1–5].

WHO and UNICEF have emphasized the first 1000 days of life when the maximal brain growth occurs, as the critical window of the period for nutritional interventions [6]. The World Health Organization (WHO), USA Maternal and Child Health Bureau and Ethiopian Federal Ministry of Health recommend exclusive breastfeeding for the first six months with early initiation, to provide colostrum, and to discourage pre-lacteal feeding and feeding by demand, at least 8 to 12 times every 24 hours for up to two years or more, and gradually supplementing with nutritionally safe, adequate, and age-appropriate complementary feeding starting at six months [2, 6–13].

Improper breastfeeding practices may significantly compromise the health, development, and survival of infants, children, and mothers. Worldwide it causes 820,000 child and 20,000 maternal deaths in a year [14]. In low- and middle-income countries, sub-optimal breastfeeding such as early introduction of complementary feeding has a high contributed to malnutrition compromising physical and cognitive development throughout life [15–17]. Over two-thirds of malnutrition is associated with improper feeding practices during the first year of life [18]. Annually, breastfeeding problems alone contributed to around 804,000 child deaths representing 12% of under-five deaths which can be prevented by optimal breastfeeding [19].

Every day, 3000–4000 infants die in the developing world from diarrhea and acute respiratory infections associated with inadequate exposure to breast milk benefits, and more than 10 million children die each year in Sub-Saharan Africa and South Asia due to poor breastfeeding practices [20]. Mainly, late initiation and nonexclusive breastfeeding are significant causes for diarrhea during the infant and young child age [21]. In Ethiopia, infants whose mothers had received low and medium BPI scores were more likely to suffer from diarrhea and other infections [22].

Breastfeeding performance index (BPI) is a scale used to quantify key breastfeeding practices into a single variable by summarizing different aspects of breastfeeding practices [23]. Though some studies have examined maternal breastfeeding practice in parts of Ethiopia [24–26], the BPI and associated maternal factors have not been well studied in Ethiopia, particularly the Northwest Amhara region. Therefore, this study aims to assess BPI scores with associated factors among mothers having infants aged 6 to 12 months in Sekela District, in Northwest Ethiopia.

## Methods and materials

### Ethics approval and consent to participate

Ethical clearance was obtained from the Ethical Review Board of Bahir Dar University College of Medicine and Health Science. The supporting letter was written from the Amhara Public Health Institute of the West Gojjam Zone Health Bureau. The zonal health bureau also wrote a permission letter to the woreda administrative and health offices. Finally supporting letters were written from woreda health offices to selected Keble administrators. Written informed consent was obtained from each study participant before the interview after explaining the purpose of the study to the respondent. Confidentiality of the information was assured and the privacy of the respondent was maintained by removing personal identities from the questionnaire. Only individuals who had consented to participate were included in the study. No one coerced study participants in any way to participate. Lastly, the participants were also informed that they have the right to abstain from the study or to withdraw at any time.

### Study setting and period

The study was conducted in the city of Gish Abay of the Sekela District, Northwest Ethiopia between April 02, 2019 and May 13, 2019. Sekela District is one of the 15 districts found in West Gojjam Zone, located 175 km and 460 km from Bahir Dar and Addis Ababa which are the capital cities of the Amhara Region and Ethiopia respectively. There are 33 Kebeles (equivalent to a county and the smallest administrative unit in Ethiopia) in the Sekela District with the total population 166,201. Among them 85,233 are female and 41,015 of whom are of reproductive age. According to the District Health Office 2011 Report, the total number of infants aged between 6 to 12 months was 3,136. There are 8 health centers and 31 health posts with 19 public health officials, 26 Midwives, 45 Nurses, and 77 Health Extension Workers [27].

**Study design and population.** A community based cross-sectional study design was used to assess the BPI and its associated factors. The study sample was obtained from all mothers who had an infant between six and twelve months of age, in randomly selected Kebeles of the Sekela District.

**Sample size determination.** The single population proportion formula was used to determine the required sample size by considering the following assumptions: prevalence of improper BPI scores was estimated at *56.9%*, based on a recently conducted study of infants aged 6 to 12 months [24]; the desired degree of precision of 5%; and using $Z\alpha/2$ as the value of the standard normal distribution corresponding to a significant level of alpha (**α**) of 0.05.

$$n = \frac{(Z_{\alpha/2})^2 P(1-P)}{d^2}$$

Due to the multistage sampling technique of the study, a design effect size of 1.5 was considered, and an adjustment for a 10% non-response rate yielded the final sample size of n = 622.

**Sampling technique and sampling procedure.** Before actual data collection, a house to house survey was done identifying 964 index mothers with an infant aged six to twelve months. Using a multi-stage sampling technique to select the study participants, ten Kebeles were randomly selected from 33 Kebeles in the district. Selected Kebeles had similar socio-demographic characteristics. From the list of survey registration numbers, individual participants from households in each of the ten sample Kebeles were randomly selected for participation in the study.

**Data collection procedure.** A structured interviewer-administered questionnaire adapted from previous publications was used to collect data [25, 28]. Questions included socio-

demographic, obstetric, and maternal health service related factors; infant related factors; and maternal BPI scores in the first six months after the birth of their infants. Breastfeeding performance index (BPI) scores assessed positive BPI scores of 1 point each for seven healthy breastfeeding practices such as early initiation, frequent, and exclusive breastfeeding until 6 months of age, and not supplementing with pre-lactael food, bottles, solids, or formula. by allocating one point for each of the following: early breastfeeding initiation; pre-lacteal feeds; bottle feeding; exclusive breastfeeding; not receiving liquids; not receiving formula or other milk, and not receiving solids (Table 1).

The BPI scores were summed to give a total score that could range between 0 and 7. The BPI scores were then classified as Low BPI (0–3), Medium BPI (4–5), or High BPI (6–7) [22]. The lowest two tertiles (low and medium) were merged as one category of the outcome variable representing poor breastfeeding practice [25, 29]. Because having low and medium PBI scores both correlate with similar nutrition and health effects, the lowest two tertiles (0–5) were analyzed as one PBI score. The odds of low and medium BPI were nearly equal in their association with increased incidence of negative health outcomes such as diarrhea, dyspnea, cough, and fever [22].

Maternal knowledge of and attitude towards breastfeeding were determined using a 16 item assessment Bloom's Taxonomy cut of point. Knowledge and attitude scores were classified as poor knowledge or negative attitude (60% or lower), medium knowledge or neutral attitude (60–79.9%), and good knowledge or positive attitude (80%or higher) respectively. [30].

*Wealth Index.* The wealth index is characterized by ownership of different types of assets in urban and rural areas. Before creating the wealth index, all variables were transformed into numerical scale values. Yes/no answers were recoded into binary variables and variables with more than 2 categories were transformed into bivariate variables. The household wealth index was estimated using principal component analysis (PCA). The objectives of a principal component analysis are to discover or reduce the dimensionality of the data set and to identify new meaningful underlying variables. The first component explains the largest principle

**Table 1. Variables and scoring system used to create breastfeeding performance index for infants age 0–6 months, 2019.**

| Variables | 0–6 months |
|---|---|
| Breastfeeding performance index components | |
| Early breastfeeding initiation | No = 0 |
| | Yes = 1 |
| Pre-lacteal feeds | No = 1 |
| | Yes = 0 |
| Bottles feeding | No = 1 |
| | Yes = 0 |
| Exclusive breastfeeding | No = 0 |
| | Yes = 1 |
| Not receiving liquids | No = 0 |
| | Yes = 1 |
| Not receiving formula or other milk | No = 0 |
| | Yes = 1 |
| Not receiving solids | No = 0 |
| | Yes = 1 |
| Range of total score | 0–7 |

proportion of the total variance, and was used as the wealth index to represent the household's wealth. After the PCA had been calculated, the index variable was divided into three categories such as: poor, medium and rich, and each household was assigned to one of these categories of household wealth index.

### Data quality assurance

Data quality was assured by using a properly designed questionnaire adapted from previous literature. Before data collection, the questionnaire was pre-tested using 5% of the total sample size from one Kebele with similar socio-demographic characteristics as the study Kebeles. Intensive training was provided for data collectors and supervisors prior to data collection. Training involved instructions on the questions to be asked, their meaning, how to ask them, and how to record the answers. Supervisors traveled with data collection teams to observe and ensure that their teams adhered to protocol. Each questionnaire was checked daily by supervisors, and principal investigators. Random re-testing of the households were done to ensure the reliability of the data.

### Data processing and analysis

Data was cleaned, coded, and entered into EpiData Version 3.1 and exported to SPSS Version 23 statistical software for analysis. Further data management (variable recoding) was done after exporting to SPSS. Data analysis included basic descriptive and analysis of potential factors associated with BPI scores. Descriptive statistics were calculated for frequencies and summarized in graphs and tables. Binary logistic regression was used for analysis. Bi-variable and multivariable logistic regression was done to show significant associations between variables and BPI scores. A p-value < 0.2 on bi-variable analyses was entered into a multivariable logistic regression model to control for confounders. The model fitness was tested using the Hosmer and Lemeshow test with the p-value set at 0.33 (p-value > 0.05). Variables with a p-value < 0.05 set at a 95% confidence interval were considered as statistically significant. The strength and direction of associations with the outcome variables were checked using the adjusted odds ratios set at a 95% confidence interval.

Household wealth status was estimated using principal component analysis (PCA). The economic variables were converted to binary variables. Those binary variable values owned by less than 5% and more than 95% of the sample were removed from analysis because they could distinguish between richer and poorer households. After the PCA was run, the index variables were divided into three categories (poor, medium, and rich) and each surveyed household was assigned to one of these household wealth categories.

## Results

### Socio-demographic characteristics of respondents

A total of 605 women having an infant aged six to twelve months representing a response rate of 97.3% were included for data analysis. More than half (58%) of respondents were between 26 and 35 years old, with the mean (SD) age of 28.54 (±4.9). Almost all (99%) respondents represented the Amhara ethnic group and all were Orthodox Christian followers. Three hundred eighteen (52.6%) were unable to read and write (Table 2).

### Maternal related characteristics of study respondents

Five hundred fifty (90.9%) study participants attended at least one antenatal care visit. From these, only 196 (35.6%) of them attended four antenatal visits. Four hundred seventy-two

**Table 2. Socio demographic characteristics of study participants in Sekela district, West Gojjam North West Ethiopia, 2019 (N = 605).**

| Variables | Frequency | Percentage |
|---|---|---|
| Age in years | | |
| 18–25 | 178 | 29.4% |
| 26–35 | 352 | 58.2% |
| ≥36 | 75 | 12.4% |
| Region) | | |
| Amhara | 598 | 98.8% |
| Others | 7 | 1.2% |
| Residence | | |
| Urban | 112 | 18.5% |
| Rural | 193 | 81.5% |
| Educational status of mother | | |
| Unable to read and write | 318 | 52.6% |
| Able to read and write | 68 | 11.2% |
| Primary level (1–8) | 146 | 24.2% |
| Secondary and preparatory | 45 | 7.4% |
| College diploma and above | 28 | 4.6% |
| Marital status | | |
| Married and Live with husband | 539 | 89.1% |
| Married and Don't Live with husband | 51 | 8.4% |
| Single (unmarried divorced &widowed) | 15 | 2.5% |
| Occupational status of mother | | |
| House wife | 495 | 81.8% |
| Government employee | 36 | 6.0% |
| Merchant | 53 | 8.8% |
| Daily labor | 21 | 3.4% |
| Wealth index of the household | | |
| Poor | 309 | 51.1% |
| Medium | 155 | 25.6% |
| Rich | 141 | 23.3% |
| Husband educational status (n = 590) | | |
| Unable to read and write | 134 | 22.7% |
| Primary level | 356 | 60.3.% |
| Secondary and preparatory level | 60 | 10.2% |
| College diploma and above | 40 | 6.8% |
| Husband occupational status (n = 590) | | |
| Farmer | 446 | 75.6% |
| Government and private sector employee | 50 | 8.5% |
| Daily labor | 32 | 5.4% |
| Merchant | 62 | 10.5% |

(78%) study participants gave birth at healthcare facilities. Only 196 (32.4%) study participants attended a postnatal care follow-up. Four hundred thirty-four (71.7%) and 411 (67.9%) of the study participants had good knowledge of, and positive attitude towards breastfeeding respectively. (Table 3).

**Table 3. Maternal health services among mother having infant 6–12 months old in Sekela District, West Gojjam zone North West Ethiopia, 2019 (N = 605).**

| Variables | | Frequency (%) |
|---|---|---|
| At least one ANC follow up (n = 605) | No | 55 (9.1) |
| | Yes | 550 (90.9) |
| Number of ANC visit (n = 550) | One visit | 30 (5.5) |
| | Two visit | 98 (17.8) |
| | Three visit | 226 (41.1) |
| | Four visit | 196 (35.6) |
| Breastfeeding advice during ANC (550) | Yes | 130 (23.6) |
| | No | 420(76.4) |
| Pregnancy intension (n = 605) | Wanted | 491 (81.2) |
| | Unwanted | 114 (18.8) |
| Place of delivery (n = 605) | Home | 133 (22.0) |
| | Health facility | 472 (78.0) |
| Mode of delivery (n = 605) | C/S delivery | 17 (2.8) |
| | Vaginal delivery | 588(97.2) |
| Post natal care follow up(n = 605) | Yes | 196 (32.4%) |
| | No | 409 (67.6%) |
| Breastfeeding advice during PNC (n = 196) | Yes | 166 (84.7%) |
| | No | 30 (15.3%) |
| Maternal knowledge (n = 605) | Good | 434 (71.7%) |
| | Medium | 129 (21.3%) |
| | Poor | 42 (7.0%) |
| Maternal attitude (n = 605) | Positive | 411 (67.9) |
| | Neutral | 105 (17.4%) |
| | Negative | 89 (14.7%) |

## Infant characteristics of study participants

Three hundred eighty-eight (64.1%) study participants had three or fewer infants. One hundred seventy-seven (42.2%) study participants had birth spacing intervals of at least three years between the last two children (Table 4).

**Table 4. Infant related factors associated with breast feeding performance index among mothers having 6–12 months infant in Sekela District, Northwest Ethiopia, 2019 (N = 605).**

| Variables | Frequency | Percentage (%) |
|---|---|---|
| Birth order (n = 605) | | |
| First-third | 388 | 64.1 |
| Fourth-fifth | 133 | 22.0 |
| Sixth and above | 84 | 13.9 |
| Birth interval (n = 419) | | |
| ≤3 years | 177 | 42.2 |
| 4–5 years | 220 | 52.5 |
| ≥6 years | 22 | 5.3 |
| Number of under five children(n = 605) | | |
| One | 381 | 63.0 |
| Two | 224 | 37.0 |

**Table 5. Breast feeding performance indicators of the study participants in Sekela district Northwest Ethiopia, 2019 (N = 605).**

| Breast feeding dimensions | Frequency | Percentage (%) | Score |
|---|---|---|---|
| Timely initiation of breastfeeding | | | |
| <1hr | 417 | 68.9 | 1 |
| >1hr | 188 | 31.1 | 0 |
| Pre-lacteal feeding | | | |
| Yes | 58 | 9.6 | 0 |
| No | 547 | 90.4 | 1 |
| Start fluid feeding before six month | | | |
| Yes | 206 | 34.0 | 0 |
| No | 499 | 66.0 | 1 |
| Start solid/semi-solid before 6 month | | | |
| Yes | 71 | 11.7 | 0 |
| No | 534 | 88.3 | 1 |
| Breast feeding until six month | | | |
| Yes | 585 | 96.7 | 1 |
| No | 20 | 3.3 | 0 |
| Avoid bottle feeding before 6 month | | | |
| Yes | 412 | 68.1 | 1 |
| No | 193 | 31.9 | 0 |
| No Formula feeding before 6 month | | | |
| Yes | 583 | 96.4 | 1 |
| No | 22 | 3.6 | 0 |
| Breast feeding performance index | | | |
| Poor | 246 | 40.7 | |
| Good | | 59.3 | |

## Maternal breastfeeding practice and breastfeeding performance index scores

Five hundred eighty-five (96.7%) study participants had breastfed their infants for at least six months. However, one hundred eighty-eight (31.1%) study participants delayed initiation of breastfeeding. Sixty-six (10.9%) study participants avoided feeding colostrum to their infant. Fifty-eight (9.6%) infants received pre-lacteal foods within three days after birth, while 193 (31.9%) infants were exposed to bottle feeding.

Moreover, 206 (34.0%) and 71(11.7%) of infants were exposed to fluids and solid or semi-solid foods before six months of age, respectively. Five hundred eighty-three (96.4%) study participants did not use a formula to feed their index infants. Two hundred thirty-nine (39.5%) infants were exposed to non-exclusive breastfeeding in the first six months after birth. The most frequently mentioned reasons for starting fluids, and solid or semi-solid foods before six months included insufficient breast milk 147 (61.5%), the mother returning to work 58 (24.3%), and other significant pressure 27 (11.3%). Three hundred fifty-nine (59.3%) study participants had high BPI scores (Table 5).

## Factors associated with breastfeeding performance index

In bivariate analysis, BPI scores were significantly associated with maternal formal education, maternal age, maternal occupation, the mother living with her husband, household wealth index, pregnancy intension, number of ANC visits, breastfeeding advice during ANC visits, delivery place, mode of delivery, PNC follow up, birth order, birth interval, number of

under-five children, breastfeeding knowledge and maternal attitude towards breastfeeding. In the final multiple logistic regressions adjusted for confounders, BPI scores were significantly associated with mother living with her husband, maternal occupation, breastfeeding knowledge, attitude towards breastfeeding, number of ANC follow ups, and postnatal follow up.

Women who were merchants were 2.8 (AOR = 2.8, 95%CI (1.05–7.15) times more likely to have low BPI scores when compared to housewives. Women who live alone were 3.2 (AOR = 3.2, 95%CI (1.15–8.82) times more likely to have a low BPI score when compared to those who live with their husbands. Women who had at least four antenatal care visits were 65% (AOR = 0.35; 95% CI (0.15–0.82), and who had third antenatal care visits were 58% (AOR = 0.42; 95%CI (0.20–0.88) times less likely to have low BPI scores as compared with those who had two or fewer antenatal care visits respectively. Women who had a history of postnatal care follow up were 65% (AOR = 0.35, 95%CI (0.19–0.64) times less likely to have a low BPI scores compared to those who had no postnatal care service. Women with a negative attitudes towards breastfeeding were 2.7 (AOR = 2.70, 95%CI (1.13–6.45) and those with neutral attitudes were 2.31 (AOR = 2.31, 95%CI (1.14–6.45) times more likely respectively, to have low BPI scores compared to those who had a positive attitude. Similarly, women with poor breast-feeding knowledge were 3.19 (AOR = 3.19, 95%CI (1.14–8.89) times more likely to have low BPI scores compared to those who had good knowledge. In addition, women who had medium breastfeeding knowledge were 2.83 (AOR = 3.19, 95%CI (1.42–5.65) times more likely to have low BPI scores when compared to women with good breastfeeding knowledge (Table 6).

## Discussion

This study was conducted to assess BPI scores with associated factors of women having infants less between 6 and 12 months of age. This study shows the prevalence of low BPI scores was 40.7%. This finding is lower than previous studies from different regions of Ethiopia such as Afar [29] and Sidama [24]. In this study, initiating fluid feeding (34.0%) was lower than in the study conducted in Mecha District West Gojjam, Ethiopia in 2012 [31], and Debre Berhan town Amhara Ethiopia in 2015 [32]. The BPI scores of this study are higher than in another African study, in Somaliland [33]. The prevalence of mothers who initiated later breastfeeding was higher than a study conducted in Debre Berhan in Amhara, Ethiopia in 2015 [32]. The current study showed that non-exclusive breastfeeding was 39.5% which is higher than a study conducted in Arba Minch, Southern Ethiopia [34]. The identified factors for low BPI score in this study were Single mothers or mothers not living with their husbands that is similar to the studies conducted in Wollo, Ethiopia in 2018 [35], Jimma Arjio, Ethiopia, in 2012 [26], and Pakistan, 2016 [36] and poor knowledge and negative attitudes about maternal breastfeeding that is consistent with previous studies in Debre Markos [37], Mecha district [31], Gonder [38] Shashemene [39], Sidama [40], Offa district South Ethiopia [41] and Gamo Goffa [42]. Whereas factors for higher BPI score were non-merchant mothers which is consistent with studies done in Debre Tabor [43], Hawassa [44], and Shashemene [39] and mothers receiving antenatal and postnatal care which is consistent with studies conducted in Mecha district [31], Hulu District, South Ethiopia [24]; Azezo District Northwest Ethiopia [45], and Southwest Somaliland [33].

The possible reason for the lower prevalence of BPI scores in the study area from other studies might be different study times, and active involvement of non-governmental organizations in those study areas. Whereas for a higher prevalence of BPI scores might be nutrition and health service coverage and socio-demographic differences. According to the Mini-Demographic and Health Survey 2019 by the Ethiopia Ministry of Health [46], four or more ANC

**Table 6. Logistic regression analysis result for factors associated with breast feeding performance index in Sekela district west Gojjam zone North West Ethiopia 2019 (N = 605).**

| Variables | Low/medium BFPI | | COR (95%) | AOR (95%) |
|---|---|---|---|---|
| | Yes | No | | |
| Maternal age (n = 605) | | | | |
| 18–25 | 80 | 98 | 1 | 1 |
| 26–35 | 117 | 235 | 0.61(0.42–0.88) | 0.66(0.20–1.22) |
| > = 36 | 49 | 26 | 2.31(1.32–4.04) | 0.74(0.14–3.28) |
| Wealth index (n = 605) | | | | |
| Poor | 159 | 150 | 1 | 1 |
| Medium | 44 | 111 | 0.37(0.25–0.57) | 0.56(0.28–1.12) |
| Rich | 43 | 98 | 0.41(0.27–0.63) | 0.60(0.30–1.20) |
| Maternal occupation | | | | |
| House wife | 187 | 308 | 1 | 1 |
| Government employee | 14 | 22 | 1.05 (0.52–2.10) | 1.28(0.24–6.72) |
| Merchant | 30 | 23 | 2.15(1.21–3.81) | 2.75(1.05–7.15) |
| daily labor | 15 | 6 | 4.12(1.57–10.80) | 1.41(0.28–7.03) |
| Mother live with her husband | | | | |
| Yes | 200 | 330 | 1 | 1 |
| No | 32 | 19 | 2.86(1.58–5.17) | 3.18(1.15–8.82) |
| Formal education | | | | |
| Yes | 75 | 144 | 0.66(0.46–0.92) | 0.54(0.24–1.23) |
| No | 171 | 215 | 1 | 1 |
| Pregnancy intension | | | | |
| Wanted | 172 | 317 | 1 | 1 |
| Unwanted | 74 | 42 | 3.25(2.13–4.95) | 1.73(0.85–3.54) |
| Birth order | | | | |
| first up to third | 140 | 248 | 1 | 1 |
| fourth and fifth | 54 | 79 | 1.21 (0.8–1.81) | 1.03(0.53–2.00) |
| sixth and above | 52 | 32 | 2.88 (1.77–4.68) | 0.89(0.34–2.45) |
| Birth interval | | | | |
| ≤3Years | 104 | 73 | 1 | 1 |
| 4-5years | 55 | 165 | 2.49(1.00–6.25) | 0.55(0.28–1.10) |
| ≥6Years | 8 | 14 | 0.58(0.23–1.47) | 1.39(0.39–5.00) |
| Having number of under 5year child | | | | |
| One | 127 | 254 | 0.58(0.23–1.47) | 0.77(0.40–1.49) |
| Two | 119 | 105 | 1 | 1 |
| Number of ANC visit | | | | |
| First and Second | 88 | 40 | 1 | 1 |
| Third visit | 81 | 145 | 0.35(0.16–0.40) | 0.42(0.20–0.88) |
| Fourth and above | 51 | 145 | 0.16(0.10–0.26) | 0.35(0.15–0.82) |
| BF advice during ANC | | | | |
| Yes | 42 | 88 | 0.58(0.43–0.98) | 0.76(0.40–1.46) |
| No | 178 | 242 | 1 | 1 |
| Place of delivery | | | | |
| Home | 74 | 59 | 2.19(1.48–3.23) | 1.81(0.94–3.49) |
| Health facility | 172 | 300 | 1 | 1 |
| Root of delivery | | | | |
| C/S | 10 | 7 | 2.13(0.08–5.68) | 2.75(0.61–12.61) |

*(Continued)*

**Table 6.** (Continued)

| Variables | Low/medium BFPI | | COR (95%) | AOR (95%) |
|---|---|---|---|---|
| | Yes | No | | |
| Vaginal | 236 | 352 | 1 | 1 |
| Post natal care | | | | |
| Yes | 46 | 150 | 0.32(0.22–0.47) | 0.35(0.19–0.64) |
| No | 200 | 209 | 1 | 1 |
| Ever informed about BF | | | | |
| Yes | 223 | 342 | 1 | 1 |
| No | 23 | 17 | 2.08(01.08–3.97) | 1.80(0.69–4.72) |
| Attitude towards to BF | | | | |
| Negative | 56 | 33 | 3.75(2.33–6.05) | 2.70(1.13–6.45) |
| Neutral | 62 | 43 | 3.19(2.05–4.06) | 2.31(1.14–4.66) |
| Positive | 128 | 283 | 1 | 1 |
| Maternal BF knowledge | | | | |
| Poor | 25 | 17 | 3.09(1.62–5.91) | 3.19(1.14–8.89) |
| Medium | 81 | 48 | 3.54(2.35–5.34) | 2.83(1.42–5.65) |
| Good | 140 | 294 | 1 | 1 |

visits and women delivering babies in facilities have increased from 32% to 43% and 26% to 48% respectively compared to the Ethiopian Demographic Health Survey 2016. In Debre Berhan and Arba Mich, the study populations were urban and included all mothers with children less than 2 years old [32, 34].

The explanations for factors associated with low BPI score in our study were it is possible that when mothers live with their husband, they get support from their husbands which decreases their workload and give them more time to breastfeed properly. Also, a mother who had poor knowledge and negative attitude about breastfeeding did not follow breastfeeding protocols like the introduction of food before six moth and early weaning breastfeeding and discontinuation of breastfeeding after six months. The possible reasons for factors associated with higher BPI score might be that housewives have more time to care for their infant and infants are not dependent on caregivers throughout the day and when mothers adhere to antenatal and postnatal care, they will get breastfeeding advice increasing their breastfeeding knowledge and motivating them to use good breastfeeding practices.

Some limitations include that data collection was based on maternal recall, and mothers may have difficulty remembering details such as the time they initiated breastfeeding. Therefore, there is a potential of recall bias which may overestimate or underestimate the results. Because the sample sizes for the low and medium BPI scores were small, it was not possible to run an ordinal logistic regression analysis, which may limit the strength of this study.

## Public health significance

Breastfeeding is an effective intervention for improving child nutrition and reducing child mortality in developing countries. In this study, the finding shows there are significant numbers of infants exposed to sub-optimal breastfeeding. These findings have implications for policies addressing problems related to breastfeeding practices in developing countries like Ethiopia. Additionally, these findings influence our understanding of breastfeeding practice interventions. Finally, the BPI may have an impact on designing outcome-oriented breastfeeding promotion interventions.

## Conclusion

The prevalence of sub-optimal breastfeeding practices in Northwest Ethiopia was very high. Marital status, maternal occupation, number of ANC visits, PNC, maternal breastfeeding knowledge, and attitude towards breastfeeding were independent predictors of low BPI scores. Nutrition promotion should be implemented by considering using evidence from this study to improve healthy breastfeeding practice.

## Supporting information

**S1 File.**
(SAV)

## Acknowledgments

We would like to offer our in-depth gratitude to the data collectors, participants for their support to us. Also would like to express our deepest appreciation and cordial thanks to Professor Barbara J. Engebretsen for here language editing of the manuscript.

## Author Contributions

**Conceptualization:** Workineh Shume Hailu, Mulat Tirfie Bayih, Netsanet Fentahun Babble.

**Data curation:** Workineh Shume Hailu.

**Formal analysis:** Workineh Shume Hailu.

**Methodology:** Mulat Tirfie Bayih, Netsanet Fentahun Babble.

**Software:** Workineh Shume Hailu.

**Writing – original draft:** Workineh Shume Hailu.

**Writing – review & editing:** Mulat Tirfie Bayih, Netsanet Fentahun Babble.

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
