## [Decision Letter · Decision Letter 0]

7 May 2020

PONE-D-20-02694

FOUR IN EVERY TEN INFANTS HAD EXPOSED TO INAPPROPRIATE BREASTFEEDING PRACTICE Northwest Ethiopia

PLOS ONE

Dear Mr Bayih,

Thank you for submitting your manuscript to PLOS ONE. After careful consideration, we feel that it has merit but does not fully meet PLOS ONE’s publication criteria as it currently stands. Therefore, we invite you to submit a revised version of the manuscript that addresses the points raised during the review process.

The manuscript has been assessed by two reviewers, their comments are appended below.

The reviewers have raised major concerns about the study, particularly regarding the language, the reporting and methodology used. They feel that the manuscript requires copyediting for English usage and grammar. In addition, they have asked for further clarification regarding the cofounders considered and the questionnaire used.

Please carefully revise the manuscript to address all comments raised.

We would appreciate receiving your revised manuscript by Jun 20 2020 11:59PM. To enhance the reproducibility of your results, we recommend that if applicable you deposit your laboratory protocols in protocols.io, where a protocol can be assigned its own identifier (DOI) such that it can be cited independently in the future. For instructions see: http://journals.plos.org/plosone/s/submission-guidelines#loc-laboratory-protocols

We look forward to receiving your revised manuscript.

Kind regards,

Sara Fuentes Perez, PhD

Staff Editor

PLOS ONE

Journal Requirements:

The name of the colleague or the details of the professional service that edited your manuscriptA copy of your manuscript showing your changes by either highlighting them or using track changes (uploaded as a *supporting information* file)

http://ijp.mums.ac.ir/pdf_6433_a4be58a59b5f56dc402986c36872b2b9.html

https://www.ncbi.nlm.nih.gov/pmc/articles/PMC4662817/

The text that needs to be addressed involves the Introduction.

In your revision ensure you cite all your sources (including your own works), and quote or rephrase any duplicated text outside the methods section. Further consideration is dependent on these concerns being addressed.

4. Please address the following:

- Please include additional information regarding the survey or questionnaire used in the study and ensure that you have provided sufficient details that others could replicate the analyses. For instance, if you developed a questionnaire as part of this study and it is not under a copyright more restrictive than CC-BY, please include a copy, in both the original language and English, as Supporting Information.

- Please ensure you have thoroughly discussed any potential limitations of this study within the Discussion section.

5. Please modify the title to ensure that it is meeting PLOS’ guidelines (https://journals.plos.org/plosone/s/submission-guidelines#loc-title). In particular, the title should be "specific, descriptive, concise, and comprehensible to readers outside the field" and in this case it is not informative and specific about your study's scope and methodology.

Additional Editor Comments (if provided):

Reviewers' comments:

Reviewer's Responses to Questions

**Comments to the Author**

1. Is the manuscript technically sound, and do the data support the conclusions?

Reviewer #1: Partly

Reviewer #2: Partly

2. Has the statistical analysis been performed appropriately and rigorously? 

Reviewer #1: Yes

Reviewer #2: Yes

3. Have the authors made all data underlying the findings in their manuscript fully available?

Reviewer #1: Yes

Reviewer #2: No

4. Is the manuscript presented in an intelligible fashion and written in standard English?

Reviewer #1: No

Reviewer #2: No

5. Review Comments to the Author

Reviewer #1: Need major improvement in English, suggest to be reviewed by a native medical writer.

The scoring or index of BFPI need to be elaborated further, specifically on the different measurement or parameter used, compare to the original source. The adaptation process of BFPI also need to be explain exhaustively.

Has the questionnaire been validated? Need to indicate if yes or not and explain the reason/argumentation. If yes what is the Cronbach score?

Important to briefly describe 'what is principal component analysis (PCA) is?' and the rational of using it is still also missing.

It is extremely obvious that most of the subject were not able to read and write, however the authors failed discussed this in depth. This is a major lack of the manuscript.

The occupational factors and other work related determinants should also be explained in detail, to draw a line with the breastfeeding performance index, even though the percentage of the subject who are worker are only around 20%.

Reviewer #2: Overall a very interesting paper looking at predictors of higher levels of breastfeeding practices in Ethiopia. This study is important since no study has been published in this specific geographic region. However, I think this paper would greatly benefit to some restructuring of the introduction and discussion. I also think the public health significance of this study needs to be further stated. Perhaps with additional analyses where they look at the impact of the breastfeeding index on infant outcomes.

Abstract

1. Please add more detail into what is meant by associated factors. Do you mean things that are associated with the index?

Introduction

1. “Breastfeeding creates an inimitable psychosocial bond between the mother and baby enhances modest cognitive development and it is the underpinning of the infant’s wellbeing in the first year of life even into the second year of life with appropriate complementary foods from 6 months[1-4].” Very long sentence and it seems like you are trying to say two important things. The benefits of breastfeeding and current recommendations. I would suggest putting these into two separate sentences.

2. You have very good and interesting information in the introduction. However, I think the introduction would benefit from some restructuring. Right now, it seems like the benefits and outcomes of breastfeeding are randomly dispersed throughoug. Instead, make sure to have all benefits in one paragraph, all recommendations in one paragraph, and do this with all major themes. That will help the transition between ideas.

Methods

1. To help readers, please state what Kebles are.

2. I am so glad you included how you determined your effect size. I would suggest using the more common terms of power. And then include what you set the main estimators to.

3. For the sampling procedure, it would be good to know if the Kebles were similar in terms of sociodemographic characteristics

4. Important to know how old the children/infants were when the interviews took place. If they were still breastfeeding at the interview, how did you deal with those participants

5. I would suggest making a figure to show exactly how the index was created. Right now, I find it a little confusing and hard to follow. So if you don’t want to make the figure I would suggest making the text a bit more clear. I think this is very important since the index is very interesting and is the major strength of your study

6. Please keep in mind that just because questionnaires were used in previous studies doesn’t mean they provide high data quality. I would add more details into how you considered if these questionnaires provided good data

7. Please also include a list of all confounders you considered

8. Did you get institutional review board approval?

Results

1. Is are any outcomes you can look at? To see if the breastfeeding index affects some of the outcomes you discussed in the introduction. This would really strengthen your paper.

Discussion

1. The discussion seems a bit long. I would like to see the first paragraph be a general overall and take home messages from your study. 2-3 discussion points for your take home messages. Strengths and limitations section. And finally conclusion with why this matters, and how you can change it..

Tables

1. Please create a table showing differences in characteristics between different levels of the breastfeeding index

Other

1. I would suggest a different title.

2. I would suggest additional editing to check for general typos

6. PLOS authors have the option to publish the peer review history of their article (what does this mean?). If published, this will include your full peer review and any attached files.

Reviewer #1: Yes: Ray Wagiu Basrowi

Reviewer #2: No

---

## [Author Response · Author response to Decision Letter 0]

29 Jul 2020

Rebuttal letter

Title: FOUR IN EVERY TEN INFANTS HAD EXPOSED TO INAPPROPRIATE BREASTFEEDING PRACTICE IN NORTHWEST ETHIOPIA 

Ms. Ref. No.: PONE-D-20-02694

We like to thank the reviewers for the positive and constructive comments with regard to the submission for publication of our manuscript in PLOS ONE. We have tried to address all comments and suggestions of the reviewers’ point by point by writing question number and answer for each and revised our manuscript accordingly. Revisions to the text were highlighted in the manuscript. 

Reviewer #1

1. Need major improvement in English, suggest to be reviewed by a native medical writer.

Answer: Thank you very much for your valuable comments. We have addressed your concerns and made correction. 

2. Scoring or index of BFPI needs to be elaborated further, specifically on the different measurement or parameter used, compare to the original source. The adaptation processes of BFPI also need to be explaining exhaustively. Has the questionnaire been validated? Need to indicate if yes or not and explain the reason/argumentation. If yes what is the Cronbach score?

Answer: Thank you very much for your valuable comments. We have addressed your concerns and made correction. The questionnaire has been using to measure breast feeding practices in Ethiopia. The pervious study was validated the breast feeding performance index questionnaire in Ethiopia. For this study, we used the already validated tools. 

3. Important to briefly describe 'what is principal component analysis (PCA) is?' and the rational of using it is still also missing.

Answer: Thank you very much for your valuable comments. We have addressed your concerns and made correction.

4. The occupational factors and other work related determinants should also be explained in detail, to draw a line with the breastfeeding performance index, even though the percentages of the subject who are worker are only around 20%.

Answer: Thank you very much for your valuable comments. We have addressed your concerns and made correction.

Reviewer #2: 

Comment: Overall a very interesting paper looking at predictors of higher levels of breastfeeding practices in Ethiopia. This study is important since no study has been published in this specific geographic region. However, I think this paper would greatly benefit to some restructuring of the introduction and discussion. I also think the public health significance of this study needs to be further stated. Perhaps with additional analyses where they look at the impact of the breastfeeding index on infant outcomes.

Answer: Thank you very much for your valuable comments. We have addressed your concerns and made correction.

Abstract

1. Please add more detail into what is meant by associated factors. Do you mean things that are associated with the index?

Answer: Thank you very much for your valuable comments. We have addressed your concerns and made correction.

Introduction

1. “Breastfeeding creates an inimitable psychosocial bond between the mother and baby enhances modest cognitive development and it is the underpinning of the infant’s wellbeing in the first year of life even into the second year of life with appropriate complementary foods from 6 months[1-4].” Very long sentence and it seems like you are trying to say two important things. The benefits of breastfeeding and current recommendations. I would suggest putting these into two separate sentences.

Answer: Thank you very much for your valuable comments. We have addressed your concerns and made correction.

2. You have very good and interesting information in the introduction. However, I think the introduction would benefit from some restructuring. Right now, it seems like the benefits and outcomes of breastfeeding are randomly dispersed throughoug. Instead, make sure to have all benefits in one paragraph, all recommendations in one paragraph, and do this with all major themes. That will help the transition between ideas.

Answer: Thank you very much for your valuable comments. We have addressed your concerns and made correction.

Methods

1. To help readers, please state what Kebeles are.

Answer: Thank you very much for your valuable comments. We have addressed your concerns and made correction.

2. I am so glad you included how you determined your effect size. I would suggest using the more common terms of power. And then include what you set the main estimators to. Answer: Thank you very much for your valuable comments. For this study we use prevalence to estimate the sample size. The study was single proportion, not comparison study. We tried to calculate the sample size by considering important factors, but, the calculated sample size was lower than the sample size estimated using prevalence. Finally, we decision to use the large sample size to increase the power of analysis and representative 

3. For the sampling procedure, it would be good to know if the Kebeles were similar in terms of socio-demographic characteristics

Answer: Thank you very much for your valuable comments. We have addressed your concerns and made correction.

4. Important to know how old the children/infants were when the interviews took place. If they were still breastfeeding at the interview, how did you deal with those participants

Answer: Thank you very much for your valuable comments. We have addressed your concerns and made correction.

5. I would suggest making a figure to show exactly how the index was created. Right now, I find it a little confusing and hard to follow. So if you don’t want to make the figure I would suggest making the text a bit more clear. I think this is very important since the index is very interesting and is the major strength of your study

Answer: Thank you very much for your valuable comments. We have addressed your concerns and made correction.

6. Please keep in mind that just because questionnaires were used in previous studies doesn’t mean they provide high data quality. I would add more details into how you considered if these questionnaires provided good data

Answer: Thank you very much for your valuable comments. We have addressed your concerns and made correction.

7. Please also include a list of all confounders you considered

Answer: Thank you very much for your valuable comments. We have addressed your concerns and made correction.

8. Did you get institutional review board approval?

Answer: Thank you very much for your valuable comments. We have addressed your concerns and made correction. The institutional review board approval protocol placed under declaration at end 

Results

1. Is/ are any outcomes you can look at? To see if the breastfeeding index affects some of the outcomes you discussed in the introduction. This would really strengthen your paper.

Answer: Thank you very much for your valuable comments. But, we did not included variables which helps to measure the effect of breastfeeding index affects 

Discussion

1. The discussion seems a bit long. I would like to see the first paragraph be a general overall and take home messages from your study. 2-3 discussion points for your take home messages. Strengths and limitations section. And finally conclusion with why this matters, and how you can change it.

Answer: Thank you very much for your valuable comments. We have addressed your concerns and made correction. 

Tables

1. Please create a table showing differences in characteristics between different levels of the breastfeeding index

Answer: Thank you very much for your valuable comments. We have addressed your concerns and made correction.

 Other

1. I would suggest additional editing to check for general typos

Answer: Thank you very much for your valuable comments. We have addressed your concerns and made correction.

---

## [Editor Report · Decision Letter 1]

12 Aug 2020

PONE-D-20-02694R1

FOUR IN EVERY TEN INFANTS IN NORTHWEST ETHIOPIA EXPOSED TO SUB-OPTIMAL BREASTFEEDING PRACTICE.

PLOS ONE

Dear Dr. Bayih,

Thank you for submitting your manuscript to PLOS ONE. After careful consideration, we feel that it has merit but does not fully meet PLOS ONE’s publication criteria as it currently stands. Therefore, we invite you to submit a revised version of the manuscript that addresses the points raised during the review process.

ACADEMIC EDITOR: Please restructure the discussion so that you compare your findings with previous literature in one paragraph; followed by two paragraphs providing an explanation into your results. Right now, you have the majority of information you need, there just needs to be some restructuring of paragraphs 2-7 of the discussion.

We look forward to receiving your revised manuscript. Further, in order to uphold the integrity of the scientific process, I think it is important for me to disclose that I participated as a reviewer for the initial evaluation of your manuscript.

Kind regards,

Jordyn Tinka Wallenborn, PhD., MPH

Academic Editor

PLOS ONE

Additional Editor Comments (if provided):

Please restructure the discussion so that you compare your findings with previous literature in one paragraph; followed by two paragraphs providing an explanation into your results. Right now, you have the majority of information you need, there just needs to be some restructuring of paragraphs 2-7 of the discussion.

---

## [Author Response · Author response to Decision Letter 1]

17 Aug 2020

Rebuttal letter

Title: FOUR IN EVERY TEN INFANTS HAD EXPOSED TO INAPPROPRIATE BREASTFEEDING PRACTICE IN NORTHWEST ETHIOPIA 

Ms. Ref. No.: PONE-D-20-02694

We have tried to address all comments and suggestions of the ACADEMIC EDITOR/ Editor Comments point by point by revised our manuscript accordingly. Revisions to the text were highlighted in the manuscript. 

ACADEMIC EDITOR/ Editor Comments: Please restructure the discussion so that you compare your findings with previous literature in one paragraph; followed by two paragraphs providing an explanation into your results. Right now, you have the majority of information you need, there just needs to be some restructuring of paragraphs 2-7 of the discussion.

Answer: Thank you very much for your valuable comments. We have addressed your concerns and restructure the discussion part.

---

## [Editor Report · Decision Letter 2]

20 Aug 2020

FOUR IN EVERY TEN INFANTS IN NORTHWEST ETHIOPIA EXPOSED TO SUB-OPTIMAL BREASTFEEDING PRACTICE.

PONE-D-20-02694R2

Dear Dr. Bayih,

We’re pleased to inform you that your manuscript has been judged scientifically suitable for publication and will be formally accepted for publication once it meets all outstanding technical requirements.

Kind regards,

Jordyn Tinka Wallenborn, PhD., MPH

Guest Editor

PLOS ONE
---

## [Editor Report · Acceptance letter]

3 Sep 2020

PONE-D-20-02694R2 

FOUR IN EVERY TEN INFANTS IN NORTHWEST ETHIOPIA EXPOSED TO SUB-OPTIMAL BREASTFEEDING PRACTICE. 

Dear Dr. Bayih:

I'm pleased to inform you that your manuscript has been deemed suitable for publication in PLOS ONE. Congratulations! Your manuscript is now with our production department. 

Kind regards, 

on behalf of

Dr. Jordyn Tinka Wallenborn 

Guest Editor

PLOS ONE